Assessing the diversity of whiteflies infesting cassava in Brazil

Xavier Cesar A.D. 1
Nogueira Angélica Maria 1
Bello Vinicius Henrique 2
Watanabe Luís Fernando Maranho 2
Barbosa Tarsiane Mara Carneiro 1
Alves Júnior Miguel 3
Barbosa Leonardo 4
Beserra-Júnior José E.A. 5
Boari Alessandra 6
Calegario Renata 7
Gorayeb Eduardo Silva 8
Honorato Júnior Jaime 9
Koch Gabriel 7
Lima Gaus Silvestre de Andrade 10
Lopes Cristian 4
de Mello Raquel Neves 11
Pantoja Késsia 6
Silva Fábio Nascimento 8
Ramos Sobrinho Roberto 10
Santana Enilton Nascimento 12
da Silva José Wilson Pereira 13
Krause-Sakate Renate renate.krause@unesp.br 2
Zerbini Francisco M. zerbini@ufv.br 1
1 Dep. de Fitopatologia/BIOAGRO, Universidade Federal de Viçosa , Viçosa , MG , Brazil
2 Dep. de Proteção Vegetal, Universidade Estadual Paulista , Botucatu , SP , Brazil
3 Faculdade de Engenharia Agronômica, Universidade Federal do Pará , Altamira , PA , Brazil
4 Instituto Federal do Sudeste de Minas Gerais , Rio Pomba , MG , Brazil
5 Dep. de Fitotecnia, Universidade Federal do Piauí , Teresina , PI , Brazil
6 Embrapa Amazônia Oriental , Belém , PA , Brazil
7 Dep. de Fitotecnia e Fitossanidade, Universidade Federal do Paraná , Curitiba , PR , Brazil
8 Centro de Ciências Agroveterinárias, Universidade do Estado de Santa Catarina , Lages , SC , Brazil
9 Centro Multidisciplinar do Campus de Barra, Universidade Federal do Oeste da Bahia , Barra , BA , Brazil
10 Centro de Ciências Agrárias/Fitossanidade, Universidade Federal de Alagoas , Rio Largo , AL , Brazil
11 Embrapa Arroz e Feijão , Santo Antonio de Goiás , GO , Brazil
12 Instituto Capixaba de Pesquisa e Extensão Rural , Linhares , ES , Brazil
13 Faculdade de Engenharia Florestal, Universidade Federal do Pará , Altamira , PA , Brazil
Gillespie Joseph
Electronic publication date: 2021 Jul 15
Publication date: 2021
Volume: 9
Electronic Location ID: e11741
Received 2021 Feb 4; Accepted 2021 Jun 17
Copyright: ©2021 Xavier et al.
Copyright year: 2021
Copyright holder: Xavier et al.
License: This is an open access article distributed under the terms of the Creative Commons Attribution License, which permits unrestricted use, distribution, reproduction and adaptation in any medium and for any purpose provided that it is properly attributed. For attribution, the original author(s), title, publication source (PeerJ) and either DOI or URL of the article must be cited.
License URL: https://creativecommons.org/licenses/by/4.0/

Keywords: Geminivirus, Virus vector, Manihot esculenta

Funding: CAPES 01 CNPq 409599/2016-6 405684/2018-5 Fapemig APQ-03276-18 FAPESP 2017/21588-7 This work was funded by CAPES (Financial code 01), CNPq (grants 409599/2016-6 to Francisco M. Zerbini and 405684/2018-5 to Renate Krause-Sakate), Fapemig (grant APQ-03276-18 to Francisco M. Zerbini), and FAPESP (grant 2017/21588-7 to Renate Krause-Sakate). The funders had no role in study design, data collection and analysis, decision to publish, or preparation of the manuscript.

==============================
Background

The necessity of a competent vector for transmission is a primary ecological factor driving the host range expansion of plant arthropod-borne viruses, with vectors playing an essential role in disease emergence. Cassava begomoviruses severely constrain cassava production in Africa. Curiously, begomoviruses have never been reported in cassava in South America, the center of origin for this crop. It has been hypothesized that the absence of a competent vector in cassava is the reason why begomoviruses have not emerged in South America.

Methods

We performed a country-wide whitefly diversity study in cassava in Brazil. Adults and/or nymphs of whiteflies were collected from sixty-six cassava fields in the main agroecological zones of the country. A total of 1,385 individuals were genotyped based on mitochondrial cytochrome oxidase I sequences.

Results

A high species richness was observed, with five previously described species and two putative new ones. The prevalent species were Tetraleurodes acaciae and Bemisia tuberculata, representing over 75% of the analyzed individuals. Although we detected, for the first time, the presence of Bemisia tabaci Middle East-Asia Minor 1 (BtMEAM1) colonizing cassava in Brazil, it was not prevalent. The species composition varied across regions, with fields in the Northeast region showing a higher diversity. These results expand our knowledge of whitefly diversity in cassava and support the hypothesis that begomovirus epidemics have not occurred in cassava in Brazil due to the absence of competent vector populations. However, they indicate an ongoing adaptation process of BtMEAM1 to cassava, increasing the likelihood of begomovirus emergence in this crop.

Introduction

Cassava (Manihot esculenta Crantz) is a perennial shrub of the Euphorbiaceae family with great economic and social importance, especially in Africa, Asia, and Latin America. Currently, cassava is the third most important source of calories after rice and corn and is a staple food for more than one billion people living mainly in developing countries (Montagnac, Davis & Tanumihardjo, 2009). Although the botanical and geographical origin of M. esculenta is still debated, studies based on genetic markers and archaeological evidence suggest that domesticated cassava originated from the wild relative progenitor M. esculenta ssp. flabellifolia in the Amazon basin, with the domestication center located at the southern border of the Amazon in Brazil (Clement et al., 2016; Leotard et al., 2009; Olsen & Schaal, 1999; Watling et al., 2018). After its introduction in west Africa by Portuguese traders during the 16th century, cassava quickly disseminated throughout tropical Africa and Asia (Carter et al., 1997). Currently, the African continent is the world’s biggest cassava producer, followed by Asia and South America (www.fao.org/faostat/en/#data/QC). Due to its high resilience to adverse environmental conditions, especially drought, high yield per unit of land and low level of management and inputs required during its life cycle, cassava is a suitable crop for poor and small farmers, partially ensuring food security in many African countries (Alves & Setter, 2004; El-Sharkawy, 2004; Gleadow, Pegg & Blomstedt, 2016).

Nevertheless, cassava may be affected by several pathogens and pests. Whiteflies (Hemiptera: Aleyrodidae) are one of the major constraints to its production in developing countries (Herrera-Campo, Hyman & Bellotti, 2011). Whiteflies comprise a diverse group of phloem-feeding insects, with more than 1,500 species assigned to 126 genera of which over 20 species have been reported to colonize cassava worldwide (Vasquez-Ordonez et al., 2015). In addition to the direct damage due to feeding in the plant phloem, whiteflies cause indirect damage by deposition of honeydew, favoring the growth of sooty mold fungi on the leaf surface, and mainly by transmission of a broad range of viruses (Navas-Castillo, Fiallo-Olivé & Sánchez-Campos, 2011). Currently, species included in the genera Aleurodicus, Aleurothrixus, Bemisia and Trialeurodes have been shown to constitute effective vectors of plant viruses classified in five families (Chandrashekar et al., 2020; Maruthi et al., 2017; Navas-Castillo, Fiallo-Olivé & Sánchez-Campos, 2011; Njoroge et al., 2017). Aleurodicus dispersus and Aleurothrixus trachoides each transmit only one virus from the genera Ipomovirus and Begomovirus, respectively, while Trialeurodes vaporariorum and T. abutilonea transmit a few viruses included in the genera Crinivirus and Torradovirus. On the other hand, the Bemisia tabaci complex comprises one of the most important group of plant virus vectors, transmitting over 450 viruses, the majority included in the genus Begomovirus (Navas-Castillo, Fiallo-Olivé & Sánchez-Campos, 2011; Zerbini et al., 2017) but including also viruses classified in the genera Carlavirus, Crinivirus, Ipomovirus, Polerovirus and Torradovirus (Ghosh et al., 2019; Gilbertson et al., 2015; Navas-Castillo, Fiallo-Olivé & Sánchez-Campos, 2011; Whitfield, Falk & Rotenberg, 2015).

Over the last decade, advances in the use of molecular markers has led to a deep reappraisal of the taxonomic status of B. tabaci (De Barro et al., 2011; Dinsdale et al., 2010). Based on molecular phylogeny of the mitochondrial cytochrome oxidase I (mtCOI) gene, it has been proposed that B. tabaci consists of a complex of more than 40 cryptic (morphologically indistinguishable) species (De Barro et al., 2011; Dinsdale et al., 2010; Lee et al., 2013; Mugerwa et al., 2018; Vyskočilová, Seal & Colvin, 2019). Partial or complete reproductive isolation and biological and ecological differences among distinct species within the complex support the proposed classification (De Marchi et al., 2017; Gilbertson et al., 2015; Malka et al., 2018; Qin, Pan & Liu, 2016). The global dissemination of polyphagous and invasive species, such as B. tabaci Middle East-Asia Minor 1 (BtMEAM1) and B. tabaci Mediterranean (BtMED), have caused major changes in the epidemiology of crop-infecting begomoviruses such as Tomato yellow leaf curl virus (TYLCV), currently present in all the main tomato producing areas of the world (Lefeuvre et al., 2010; Mabvakure et al., 2016; Pan et al., 2012). In addition, the dissemination of polyphagous whiteflies has favored the transfer of indigenous begomoviruses from wild reservoir hosts to cultivated plants, as occurred in tomato crops in Brazil after the introduction of BtMEAM1 in the mid-1990’s (Ribeiro et al., 1998; Rocha et al., 2013).

Specific associations between endemic populations of B. tabaci and indigenous begomoviruses have also led to the emergence of severe epidemics in crops (Fauquet & Fargette, 1990; Pan et al., 2018). Cassava mosaic disease (CMD) is considered the most significant constraint to cassava production in Africa (Jacobson, Duffy & Sseruwagi, 2018); (Rey & Vanderschuren, 2017) and has been expanding to south and southeast Asia in recent years (Minato et al., 2019; Wang et al., 2019; Wang et al., 2016). CMD is caused by viruses of the genus Begomovirus (family Geminiviridae), which are transmitted in a circulative manner by whiteflies of the Bemisia tabaci cryptic species complex (Zerbini et al., 2017). Also, because cassava can be vegetatively propagated, transmission by infected root stock is important for the long-distance spread of CMD (Wang et al., 2020). To date, nine cassava mosaic begomoviruses (CMBs) have been reported in association with CMD, seven of them in Africa and two in the Indian subcontinent (Jacobson, Duffy & Sseruwagi, 2018; Legg et al., 2015; Patil & Fauquet, 2009). The emergence of CMD seems to have been the result of the transfer of indigenous begomoviruses from wild reservoir hosts to cassava, probably mediated by endemic populations of B. tabaci that have adapted to feed in cassava since its introduction from South America (Fauquet & Fargette, 1990; Legg & Fauquet, 2004). In Africa, CMD is transmitted primarily by B. tabaci Sub-Saharan Africa (SSA1-3, SSA6 and SSA9) and B. tabaci Indian Ocean (IO) (Jacobson, Duffy & Sseruwagi, 2018; Kunz et al., 2019). B. tabaci MEAM1 has been detected in some countries and is associated mostly with the transmission of begomoviruses in tomato (Jacobson, Duffy & Sseruwagi, 2018). Although wild reservoir hosts and a possible ancestral progenitor of current begomoviruses causing CMD have not been found, the absence of cassava-infecting begomoviruses in the Americas supports an African origin for those viruses, and the presence of cassava-adapted B. tabaci species being restricted to Africa reinforces that hypothesis. The high species diversity and high level of molecular variation observed in viral populations causing CMD strongly suggests Africa as a diversification center for CMBs, with distinct CMBs recurrently emerging and evolving for a long time (De Bruyn et al., 2016; De Bruyn et al., 2012; Ndunguru et al., 2005; Tiendrebeogo et al., 2012).

Even if CMD in Africa is caused by indigenous viruses, the fact that cassava in South America has not been affected by begomoviruses is puzzling (Carabali, Belloti & Montoya-Lerma, 2010; Carabali, Montoya-Lerma & Bellotti, 2008; Patil & Fauquet, 2009). Carabali et al. (2005) suggested that the absence of cassava-infecting begomoviruses in the Americas would be due to lack of competent B. tabaci species that efficiently colonize cassava (Fauquet & Fargette, 1990; Legg & Fauquet, 2004). In Colombia, the inability of B. tabaci MEAM1 to colonize M. esculenta efficiently has been demonstrated under experimental conditions, reinforcing the above hypothesis (Carabali, Belloti & Montoya-Lerma, 2010). In addition, a recent study also from Colombia failed to detect any whitefly species of the B. tabaci complex in cassava (Gómez-Díaz, Montoya-Lerma & Muñoz Valencia, 2019).

Although whitefly diversity in Brazil has been surveyed extensively in recent years (Barbosa et al., 2015; Marubayashi et al., 2014; Marubayashi et al., 2013; Moraes et al., 2017; Moraes et al., 2018; Rocha et al., 2011), no study has been carried out specifically to explore the composition of whitefly communities colonizing cassava. Those studies carried out in other crops demonstrated that B. tabaci MEAM1 is the predominant species across Brazil in crops such as common bean, cotton, pepper, tomato and soybean. Furthermore, B. tabaci MED, which was recently introduced in Brazil, has quickly spread and currently is present in five states from the South and Southeast regions (Barbosa et al., 2015; Moraes et al., 2017; Moraes et al., 2018). A small number of whitefly samples from cassava were analyzed in those studies, with B. tuberculata and Tetraleurodes acaciae prevalent and detected exclusively in cassava (Marubayashi et al., 2014; Moraes et al., 2018). A large survey addressing whitefly diversity in cassava in its domestication center could provide clues to understand the absence of a CMD-like disease in the Americas. Moreover, this knowledge would be useful to anticipate the potential of emergence of begomoviruses in the crop and to help anticipate a management strategy.

Given this context, the objective of this work was to evaluate whitefly diversity in cassava across Brazil to infer about the absence of begomovirus occurrence in cassava. Our results demonstrated that the most prevalent species in cassava were T. acaciae and B. tuberculata. In addition, we detected for the first time the presence of BtMEAM1 colonizing cassava in Brazil. The possible implications of these findings are discussed considering the absence of CMD and the potential for its emergence in cassava fields in Brazil.

Materials & Methods

Whitefly and cassava samples

Whiteflies were collected exclusively from cassava (M. esculenta) plants across 12 Brazilian states representative of the five macroregions (North, Northeast, Midwest, Southeast and South; Fig. 1) between March 2016 and February 2019 (Table 1). To gather evidence of whether a given species was colonizing cassava, adults and nymphs from the same field were collected whenever possible (Table 1). Samples were obtained from commercial and non-commercial (subsistence) crops. Whitefly adults were sampled using a hand-held aspirator and nymphs were collected with the aid of a needle. Insects were preserved in 95% ethanol and stored at −20 °C until being used for molecular identification of the species.

Figure 1 Overview of the survey locations and whitefly species detected.

(A) Clockwise from top-left: adults and nymphs of Bemisia tuberculata colonizing cassava in Mogi Mirim, São Paulo state. Growth of sooty mould fungus on the leaf surface due to the deposition of honeydew by whiteflies. Presence of begomovirus-infected Blainvillea rhomboidea (family Asteraceae) in a cassava field in Minas Gerais state. Presence of begomovirus-infected Euphorbia heterophylla (family Euphorbiaceae) in a cassava field in Minas Gerais state. (B) Map of Brazil showing the locations where whiteflies samples were collected. The map is colored according to the regions as indicated in the legend. Blacks dots correspond to the sampled sites. Scale bar is only for Brazil map. (C) Number of adults and nymphs analyzed from each sampled site according to state. (D) Specie distribution according to region. AL, Alagoas; BA, Bahia; DF, Distrito Federal; ES, Espírito Santo; GO, Goiás; MG, Minas Gerais; MT, Mato Grosso; PA, Pará; Piauí; PR, Paraná; SC, Santa Catarina; SP, São Paulo.

Table 1 Sampled sites and whiteflies species detected in cassava in Brazil.

Sample	Date of collection	Location	Region	Geographical coordinates	Alt.	Number of samples	Whiteflies speciesc	Reference	
				Latitude	Longitude		Adb	Ny	T			
AL1	July 2018	Arapiraca, ALa	Northeast	09°48′27.66″S	36°36′40.68″W	241	15	15	30	Btu, Ta	This study	
AL2	April 2018	TeotonioVilela, AL	Northeast	09°53′16.86″S	36°23′05.52″W	167	15	15	30	BtM, BtNW, Btu, Ta	This study	
AL3	April 2018	União dos Palmares, AL	Northeast	09°11′51.84″S	36°01′51.90″W	154	14	10	24	BtM, BtNW, Btu, Ta	This study	
AL4	April 2018	Arapiraca, AL	Northeast	09°47′29.46″S	36°25′36.00″W	143	14	15	29	BtM, Btu, Ta	This study	
AL5	April 2018	Arapiraca, AL	Northeast	09°43′09.30″S	36°40′45.60″W	298	16	15	31	BtM, Ta	This study	
BA1	December 2017	Barra, BA	Northeast	11°20′52.37″S	43°12′57.44″W	409	28	0	28	BtM, Btu, Ta	This study	
BA2	March 2018	Luis Eduardo Magalhães, BA	Northeast	12°20′10.00″S	45°49′12.00″W	744	10	10	20	BtM, Btu	This study	
BA3	June 2018	Wanderley, BA	Northeast	12°13′42.04″S	43°55′50.00″W	875	14	0	14	BtM, Ta	This study	
BA4	March 2018	Cristópolis, BA	Northeast	12°13′56.08″S	44°23′11.50″W	694	13	0	13	BtM, Ta	This study	
DF1	March 2018	Planaltina, DF	Midwest	15°30′59.00″S	47°16′09.00″W	809	10	0	10	BtM	This study	
DF2	March 2018	Planaltina, DF	Midwest	15°28′57.00″S	47°20′06.00″W	757	10	0	10	BtM	This study	
DF3	March 2018	Planaltina, DF	Midwest	15°31′17.00″S	47°21′22.00″W	874	10	0	10	BtM	This study	
ES1	January 2018	Sooretama, ES	Southeast	19°06′52.02″S	40°04′46.30″W	60	11	11	22	Ta	This study	
ES2	January 2018	Marilândia, ES	Southeast	19°24′22.04″S	40°32′21.70″W	111	14	15	29	BtM, Ta	This study	
ES3	January 2018	Pinheiros, ES	Southeast	18°40′83.20″S	40°28′63.40″W	88	15	15	30	BtM, Btu, Ta	This study	
GO1	March 2018	Bela Vista, GO	Midwest	16°59′50.00″S	48°57′56.00″W	808	10	11	21	BtM	This study	
GO2	March 2018	Itaberaí, GO	Midwest	15°56′58.00″S	49°47′07.00″W	765	15	15	30	BtM, Btu	This study	
MG1	February 2018	Ouro Fino, MG	Southeast	22°16′44.00″S	46°29′33.00″W	846	10	10	20	Ta	This study	
MG2	February 2018	Pouso Alegre, MG	Southeast	22°15′01.00″S	46°58′31.00″W	677	15	15	30	BtM, Btu, Ta	This study	
MG3	February 2018	Careaçu, MG	Southeast	22°04′37.00″S	45°41′49.00″W	813	15	15	30	Btu, Ta	This study	
MG4	February 2018	Lambari, MG	Southeast	21°56′05.00″S	45°15′49.00″W	864	15	15	30	BtM, Ta	This study	
MG5	February 2018	Lima Duarte, MG	Southeast	21°50′35.00″S	43°47′01.00″W	751	11	11	22	Ta	This study	
MG6	February 2018	Rio Pomba, MG	Southeast	21°15′50.00″S	43°09′59.00″W	504	15	15	30	Btu, WtNEW2	This study	
MG7	March 2018	Florestal, MG	Southeast	19°54′13.00″S	44°25′48.00″W	793	10	10	20	Ta	This study	
MG8	March 2018	Florestal, MG	Southeast	19°51′20.00″S	44°23′58.00″W	704	10	10	20	Ta	This study	
MG9	March 2018	Florestal, MG	Southeast	19°53′39.00″S	44°24′55.00″W	769	15	15	30	BtM, Ta	This study	
MG10	March 2018	Florestal, MG	Southeast	19°52′38.00″S	44°25′21.00″W	780	16	13	29	BtM, Btu, Ta	This study	
MG11	March 2018	Florestal, MG	Southeast	19°52′38.00″S	44°25′21.00″W	780	15	14	29	Ta	This study	
MG12	March 2018	Divinópolis, MG	Southeast	20°06′21.00″S	44°55′36.00″W	782	15	15	30	Btu, Ta	This study	
MG13	March 2018	Viçosa, MG	Southeast	20°46′06.00″S	42°52′14.00″W	661	14	11	25	BtM, BtNW, Btu, Ta	This study	
MG14	April 2018	Piraúba, MG	Southeast	21°16′22.71″S	43°02′31.28″W	366	10	0	10	Ta	This study	
MG15	May 2018	Descoberto, MG	Southeast	21°28′06.28″S	42°58′05.53″W	351	15	15	30	Btu, Ta	This study	
MG16	June 2018	Mar de Espanha, MG	Southeast	21°46′07.26″S	43°04′26.62″W	508	15	15	30	Btu, Ta	This study	
MG17	July 2018	Leopoldina, MG	Southeast	21°33′50.15″S	42°40′42.97″W	254	9	0	9	Ta	This study	
MG18	August 2018	Dona Euzébia, MG	Southeast	21°19′18.83″S	42°48′37.07″W	246	17	0	17	Btu, Ta	This study	
MG19	May 2018	Caparaó, MG	Southeast	20°31′48.00″S	41°54′00.36″W	864	12	10	22	Ta	This study	
MT1	December 2017	Canarana, MT	Midwest	13°31′16.00″S	52°25′03.00″W	312	15	16	31	WtNEW1	This study	
MT2	December 2017	Canarana, MT	Midwest	13°33′47.00″S	52°15′53.00″W	415	0	20	20	Btu	This study	
MT4	January 2018	Pedra Preta, MT	Midwest	16°38′29.00″S	54°25′41.00″W	241	10	10	20	Btu	This study	
MT5	January 2018	Pedra Preta, MT	Midwest	16°39′07.00″S	54°22′27.00″W	255	10	10	20	BtM, Btu	This study	
MT6	January 2018	Pedra Preta, MT	Midwest	16°39′28.00″S	54°20′20.00″W	303	10	10	20	BtM, Btu	This study	
PA1	January 2018	Brasil Novo, PA	North	03°12′23.07″S	52°30′13.80″W	170	11	10	21	Ta	This study	
PA2	January 2018	Vitória doXingu, PA	North	03°04′51.01″S	52°10′08.80″W	130	10	12	22	Ta	This study	
PA3	January 2018	Altamira, PA	North	03°18′15.03″S	52°07′26.00″W	176	15	15	30	Btu, Ta	This study	
PA4	January 2018	Altamira, PA	North	03°09′14.60″S	52°07′49.00″W	128	15	16	31	Btu, Ta	This study	
PA5	August 2018	Belém, PA	North	01°18′20.00″S	48°26′46.00″W	46	10	10	20	BtM	This study	
PI1	April 2018	Picos, PI	Northeast	07°04′79.50″S	41°25′57.80″W	214	0	29	29	Btu, Ta	This study	
PI2	May 2018	Teresina, PI	Northeast	05°02′41.94″S	42°47′18.84″W	71	0	27	27	Btu, Ta	This study	
PR1	March 2018	Santo Antônio do Caiuá, PR	South	22°41′07.00″S	52°19′06.00″W	342	30	0	30	Btu, Ta	This study	
PR2	March 2018	Santo Antônio do Caiuá, PR	South	22°49′03.00″S	52°21′25.00″W	439	30	0	30	BtM, Btu	This study	
PR3	March 2018	Paranavaí, PR	South	23°06′03.29″S	52°29′10.00″W	471	26	0	26	Btu, Ta	This study	
PR4	March 2018	Sertanópolis, PR	South	23°02′54.00″S	50°59′54.00″W	343	10	10	20	BtM, BtMED	This study	
SC2	March 2018	Agrônomica, SC	South	27°34′40.00″S	48°32′08.00″W	34	10	0	10	BtM	This study	
SP1	July 2016	Holambra, SP	Southeast	22°36′26.00″S	47°02′50.00″W	634	10	0	10	BtMED	Moraes et al. (2018)	
SP2	July 2016	Mogi Mirim, SP	Southeast	22°28′08.00″S	46°56′25.00″W	652	10	0	10	Btu	Moraes et al. (2018)	
SP3	July 2016	Mogi Mirim, SP	Southeast	22°24′59.00″S	46°59′19.00″W	678	10	0	10	Btu	Moraes et al. (2018)	
SP4	July 2016	Mogi Mirim, SP	Southeast	22°26′44.00″S	47°04′11.00″W	689	10	0	10	Btu	Moraes et al. (2018)	
SP5	July 2016	Mogi Mirim, SP	Southeast	22°27′05.00″S	47°04′56.00″W	710	10	0	10	Btu	Moraes et al. (2018)	
SP6	July 2016	São Pedro, SP	Southeast	22°34′08.00″S	48°05′22.00″W	525	10	0	10	Btu	Moraes et al. (2018)	
SP7	July 2017	Montalvão, SP	Southeast	22°02′23.00″S	51°19′53.00″W	404	10	0	10	Btu	Moraes et al. (2018)	
SP8	July 2017	Pindamonhangaba, SP	Southeast	22°56′05.00″S	45°26′25.00″W	562	10	0	10	BtM	Moraes et al. (2018)	
SP9	March 2016	Casa Branca, SP	Southeast	21°11′32.00″S	47°48′44.00″W	598	10	0	10	BtM	Moraes et al. (2018)	
SP10	January 2019	São Pedro do Turvo, SP	Southeast	22°36′32.00″S	49°45′29.00″W	563	4	10	14	BtM, Btu, BtMED	This study	
SP11	January 2019	Oleo, SP	Southeast	22°56′32.00″S	49°26′15.00″W	584	10	0	10	BtMED	This study	
SP12	February 2019	Mogi Mirim, SP	Southeast	22°28′32.30″S	47°00′47.60″W	675	5	5	10	Btu	This study	
SP13	July 2016	Casa Branca, SP	Southeast	21°49′08.00″S	46°58′23.00″W	612	10	0	10	Btu	Moraes et al. (2018)	
Total							819	566	1385			
Notes.

a Brazilian states where samples were collected: AL, Alagoas; BA, Bahia; DF, Distrito Federal; ES, Espírito Santo; GO, Góias; MG, Minas Gerais; MT, Mato Grosso; PA, Pará; PI, Piauí; PR, Paraná; SC, Santa Catarina; SP, São Paulo.

b Ad, Adults; Ny, Nymphs; T, Total.

c Btu, Bemisia tuberculata; BtM, Bemisia tabaci MEAM1; BtMED, B. tabaci MED; BtNW, B. tabaci New World; Ta, Tetraleuroides acaciae; WtNEW1, whitefly new species 1; WtNEW2, whitefly new species 2.

To verify the presence of begomoviruses infecting cassava, foliar samples were also collected at some sampled sites (Table S1). The samples were collected randomly regardless of the presence of virus-like symptoms. The leaves were press-dried and stored at room temperature as herbarium-like samples until being used for DNA extraction.

Whitefly species identification

Whitefly species were identified by sequencing of a mtCOI fragment, as previously described (Moraes et al., 2018). When enough adults and nymphs were collected at a given sampled site, ten individuals from each stage were analyzed, and when only one stage was obtained, 20 individuals were tested (Table 1). An initial assessment of whitely diversity was done using PCR-RFLP of the mtCOI fragment. When variation in the RFLP pattern was observed in the first screening, suggesting that more than one species could be present in that site, approximately five additional individuals for each stage were sequenced according to sample availability.

Total DNA was extracted from single individual whiteflies following a Chelex protocol (Walsh, Metzger & Higuchi, 1991). Briefly, adults or nymphs were ground in 30 µl of Chelex buffer (5% Chelex in 1×Tris-EDTA) using a toothpick in a 600 µl tube. Samples were vortexed for 30 s and incubated at 99 °C for 8 min in a PTC-100 thermocycler (MJ Research). Next, the tubes were centrifuged at 14,000 g for 5 min and 20 µl of the supernatant was collected and transferred to a new tube. One microliter of the supernatant was used as a template for PCR amplification of a 800 bp fragment of the mtCOI gene using primers C1-J-2195 and L2-N-3014 (Frohlich et al., 1999; Simon et al., 1994). PCR was performed using 0.2 µM of forward and reverse primers in a final volume of 25 µl using GoTaq Colorless Master Mix (Promega), following the manufacturer’s instructions. The PCR cycles consisted of an initial denaturing step at 95 °C for 5 min, followed by 35 cycles at 95 °C for 30 s, 42 °C for 45 s and 72 °C for 1 min, with a final extension at 72 °C for 10 min. Amplified products were visualized in 0.8% agarose gels stained with ethidium bromide and directly used for RFLP analysis (Bosco et al., 2006).

RFLP analysis of the amplicons consisted of 5 µl of each PCR product digested with 0.1 unit of Taq I (Promega) in a final volume of 20 µl. Reactions were performed at 65 °C for 2 h and visualized in 1.2% agarose gels stained with ethidium bromide. To verify whether the predicted mtCOI restriction pattern corresponded to a given species according to in silico prediction, a subset of PCR products from adults and nymphs representative of distinct patterns from different sampled sites were selected and sequenced. PCR products were precipitated with 100% ethanol and 3 M sodium acetate pH 5.2 (Sambrook & Russel, 2001) and sequenced commercially (Macrogen Inc.) in both directions using primers C1-J-2195/L2-N-3014.

For a small subset of samples that failed to yield a PCR product using primers C1-J-2195 and L2-N-3014, a second screening, using a recently described primer set with improved specificity for species of the B. tabaci complex and B. afer (2195Bt and C012/Bt-sh2), was performed (Mugerwa et al., 2018). Samples that still failed to amplify or had unexpected RFLP pattern were analyzed with specific primers for T. vaporariorum (TvapF and Wfrev) (Scott et al., 2007).

Sequence comparisons and phylogenetic analysis

Nucleotide sequences were first checked for quality and assembled using Geneious v. 8.1 (Kearse et al., 2012). mtCOI sequences were initially analyzed with the BLASTn algorithm (Altschul et al., 1990) to determine the whitefly species with which they shared greatest similarity. Pairwise comparisons between all mtCOI sequences obtained here and those with higher similarities (as determined by the BLASTn search) were performed with the program SDT v. 1.2 (Muhire, Varsani & Martin, 2014) using the MUSCLE alignment option (Edgar, 2004).

For phylogenetic analyses, the final dataset was composed of 142 sequences: 95 obtained in this work and 47 sequences representative of species in the family Aleyrodidae. Sequences were retrieved from GenBank and from the updated mtCOI reference dataset for species of the Bemisia tabaci complex (Boykin, Savill & De Barro, 2017). Multiple sequence alignments were prepared using the MUSCLE option in MEGA7 (Kumar, Stecher & Tamura, 2016). Alignments were checked and manually adjusted when necessary. Phylogenetic trees were constructed using Bayesian inference performed with MrBayes v. 3.0b4 (Ronquist & Huelsenbeck, 2003). The program MrModeltest v. 2.2 (Nylander, 2004) was used to select the nucleotide substitution model with the best fit in the Akaike Information Criterion (AIC). The analyses were carried out running 50,000,000 generations with sampling at every 1,000 generations and a burn-in of 25%. The convergence was assumed when average standard deviation of split frequencies was lower than 0.001. Trees were visualized and edited using FigTree (tree.bio.ed.ac.uk/software/figtree) and CorelDRAW X5, respectively.

Virus detection in foliar samples

Total DNA was extracted as described (Doyle & Doyle, 1987) and used as a template for PCR using the DNA-A universal primer pair PAL1v1978 and PAR1c496 (Rojas et al., 1993). PCR was performed in a final volume of 25 µl using Taq DNA Polymerase (Invitrogen) following the manufacturer’s instructions. The PCR cycles consisted of an initial denaturing step at 95 °C for 5 min, followed by 35 cycles at 95 °C for 1 min, 52 °C for 1 min and 72 °C for 1 min, with a final extension at 72 °C for 10 min. PCR products were visualized in 0.8% agarose gels stained with ethidium bromide. In addition, rolling-circle amplification (Inoue-Nagata et al., 2004) followed by digestion with MspI was performed in a subset of the samples.

Diversity index and statical analysis

Simpson’s index of diversity (1-D) was calculated to verify if there was any difference in whitefly diversity across macroregions. This index represents the probability that two randomly chosen individuals in a given sampled site will belong to distinct species (Morris et al., 2014). Simpson’s index was chosen as its value increases with increasing diversity and assigns more weight to more abundant species in a sample. We assume that species colonizing cassava will be in abundance, whereas rare species that briefly visit the plant without colonizing it will be underrepresented. Simpson’s index was calculated for each sampled site separately and then pooled according to macroregions. To assess the statistical significance of the differences in diversity among regions, the non-parametric Kruskal-Wallis test followed by post hoc multiple comparison test using Fisher’s least significant difference was calculated, using the function kruskal implemented in the Agricolae package in R software (R Core Team, 2017). Non-parametric Spearman’s rank correlation coefficient analysis was performed using the ggpubr package in R software (R Core Team, 2017).

Results

High whitefly species richness in cassava in Brazil

To verify the composition of whitefly communities colonizing cassava in Brazil, sampling was performed across the country, including the main agroecological zones. A total of 66 sites from 12 states were sampled (Fig. 1; Table 1). Out of 1,385 individuals submitted to PCR-RFLP analysis, 58 adults and 37 nymphs from different locations and representing distinct restriction patterns were sequenced (Table S2). The combination of PCR-RFLP followed by sequencing showed reliability and consistency for species identification without misidentification due to incongruence between the two methods.

Based on pairwise comparisons and molecular phylogeny of the partial mtCOI gene, we identified the presence of at least seven species comprising the whitefly community in cassava (Fig. 2; Table 1). Among them, T. acaciae and B. tuberculata, both previously reported in this crop, were the most prevalent, representing over 75% of the analyzed individuals. In addition, based on the criterion of 3.5% divergence to differentiate species within the B. tabaci complex, three B. tabaci species were identified, with BtMED previously reported, and BtMEAM1 and BtNW identified for the first time in cassava fields in Brazil (Fig. 2; Table 1). The species BtMEAM1 represented 18% of the total individuals analyzed, followed by BtMED (1.6%) and BtNW (0.21%).

Figure 2 Phylogenetic relationships among whitely species, including the ones detected in cassava in Brazil, based on the sequence of the mtCOI gene.

Bayesian phylogenetic tree based on partial nucleotide sequences of the mitochondrial cytochrome oxidase (mtCOI) gene of representative individuals of each whitefly species detected in this study and reference sequences retrieved from GenBank. The tree was rooted with the aphid Aphis gossypii. Bayesian posterior probabilities are shown at the nodes. The scale bar represents the number of nucleotide substitutions per site. Nodes with posterior probability values between 0.60 and 0.80 are indicated by empty circles and nodes with values equal to or greater than 0.81 are indicated by filled circles. Clades highlighted with different colors indicate the species detected in this study. Branches highlighted in red indicate the putative new species detected here.

Furthermore, two putative new species were identified (Fig. 2), provisionally named whitefly new species 1 and 2 (WtNEW1 and WtNEW2). The WtNEW1 mtCOI sequence (KY249522) showed highest identity (80.65%) and clustered close to the T. acaciae clade, comprised of individuals reported here and three other previously reported sequences from cassava in Brazil (Fig. 2). For WtNEW2, two mtCOI sequences obtained from an adult (JX678666) and a nymph (DQ989531) shared 97.81% among them and showed highest identity with B. tabaci (adult: 82.11%; nymph: 81.68%) and clustered as a basal sister clade to the genus Bemisia (Fig. 2). Although whitefly taxonomy is predominantly based on puparial characters (Hodges & Evans, 2005) and there is no taxonomic criterion established based in mtCOI sequences for most of the groups, as has been proposed for the B. tabaci complex, the level of divergence between the two proposed new species with the closest species is similar to the level of divergence observed between species already described within the Aleyrodidae, as demonstrated in pairwise comparisons (Table S3) and phylogenetic analysis (Fig. 2). Nevertheless, further molecular and morphological characterization should be performed. Together, these results indicate the existence of a high whitefly species richness in cassava in Brazil.

Both the prevalence and the capacity to colonize cassava differ among species

Nymphs were collected for samples identified as T. acaciae, B. tuberculata, BtMEAM1 and the two new putative species (Fig. 3A), suggesting that these species may colonize cassava. Nymphs were not obtained at two sites where BtMED was prevalent (SP1 and SP12). Although it could be suggested that this species has the potential to colonize cassava due to the high prevalence of adults at these two sites, the lack of nymphs suggests otherwise. Moreover, at the sites PR4 and MT6, BtMEAM1 predominated among adults but 100% of the nymphs were B. tuberculata, suggesting that the predominance at one stage does not necessarily mean predominance in another stage. Indeed, correlation analysis between the number of adults and nymphs, performed for all sites where both stages were sampled, showed no significant correlation between them (Fig. S1). Further sampling in those sites or free-choice experiments are necessary to confirm the potential of BtMED to colonize cassava. Considering the whole sampling, we detected only three adults of BtNW, suggesting an inability of this specie to colonize cassava.

Figure 3 Composition of whitefly populations colonizing cassava in Brazil.

(A) Species composition at each sampled site according to stage of development (adult and nymphs). Asterisks indicate that nymphs were not detected. (B, C, D) Species distribution of the 1,385 individuals genotyped in this study considering the samples from all sites (B) or only samples from sites where both adults and nymphs were sampled (C) or without samples from Minas Gerais state (D).

To verify if prevalence differs among species across distinct developmental stages, the data were separated according to stage and the proportions of individuals were compared for the three most abundant species (Figs. 3B, 3C). Considering the entire data set, T. acaciae was the prevalent species, followed by B. tuberculata and BtMEAM1 (x22=152.63, P < 2.2 ×10−16). The same was true according to stage, either adults (x22=28.61, P < 6.1 ×10−07) or nymphs (x22=169.44, P < 2.2 × 10−16; Fig. 3B). However, caution is needed to interpret these results as only adults were sampled at some sites where BtMEAM1 and B. tuberculata were prevalent (Fig. 3A), which could bias the analysis, causing an underestimation of the number of nymphs for those species. Therefore, we also analyzed the data considering only those sites where both adults and nymphs were obtained. Again, T. acaciae was the predominant species followed by B. tuberculata and BtMEAM1 considering either the entire data set (x22=258.61, P < 2.2 ×10−16) or only nymphs (x22=164.47, P < 2.2 ×10−16). When only adults were considered, T. acaciae was still predominant (x22=113.52, P < 2.2 ×10−16) but no difference between B. tuberculata and BtMEAM1 was observed (x12=0.505, P = 0.477; Fig. 3C). Moreover, it could be argued that samples from Minas Gerais (MG) were overrepresented in our sampling (Fig. 1C), which could also bias the results presented above due to the predominance of T. acaciae in this state (Fig. 3A). To test this possibility, we analyzed the data excluding the samples from MG. In this case, when both stages were considered, B. tuberculata was predominant (x22=62.09, P = 3.3 ×10−14) but no difference between T. acaciae and BtMEAM1 was observed (x12=1.91, P = 0.166). When each stage was considered separately, B. tuberculata was predominant followed by Bt MEAM1 and T. acaciae (adults: x22=43.94, P = 2.9 ×10−10; nymphs: x22=84.19, P < 2.2 ×10−16). Together, these results indicate that the potential to colonize cassava differs among species, which could be due either to lower preference for the plant or to differences in the competitive ability among species during cassava colonization. In addition, they reinforce the low efficiency of BtMEAM1 to colonize cassava.

Competitive interference does not explain the differences in prevalence

Interestingly, at least two species were detected co-occurring at 51% of the sampled sites (Fig. 3A). To verify the possibility of competition among T. acaciae, B. tuberculata and BtMEAM1 to explain the observed differences in prevalence (instead of differences in host preference), the competitive capacity of these three species was inferred based on the analysis of predominance at the sites where they occurred together. Initially, we verified if there were any differences in incidence, defined here as the number of sampled sites where at least one individual belonging to one of the three species was detected (Fig. 4A). The results demonstrate that there were no differences in incidence among them (x22=1.25, P = 0.537; Fig. 4A). In addition, no differences were observed when the proportion of sites where whitefly species occurred alone or in different combinations was compared (x62=3.26, P = 0.776; Fig. 4B). However, when we compared the occurrence between BtMEAM1 and non-B. tabaci species at the sites where they occur alone, the number of sites with non-B. tabaci species was higher (x12=6.53, P = 0.011; Fig. 4B). Thus, the competitive capacity was inferred based on the proportion of individuals from each species at the fields where these species were detected co-occurring in different combinations (Fig. 4C). Interestingly, at the sites where BtMEAM1 and B. tuberculata were sampled together, B. tuberculata predominated over Bt MEAM1, suggesting higher competitive potential (Fig. 4C). For all other species combinations, no evidence of differences in competitive capacity were observed (Fig. 4C). Together, these results suggest that, rather than competition, lower host preference by BtMEAM1 explains its non-prevalence compared to T. acaciae and B. tuberculata, resulting in low colonization rate as indicated by the low number of BtMEAM1 nymphs detected in cassava (Fig. 3).

Figure 4 Incidence of Trialeurodes acaciae, Bemisia tuberculata and B. tabaci MEAM1 in cassava fields in Brazil.

(A) Incidence of Trialeurodes acaciae, Bemisia tuberculata and B. tabaci MEAM1 in cassava fields in Brazil, measured as the percentage of sampled sites where at least one individual belonging to each one of the three species was detected. Other species detected at low incidence are shown together as ”others”. (B) Venn diagram showing the proportion of sites where each one of the three whitefly species occur alone or in different combinations. (C) Competitive capacity inferred based on the prevalence of individuals from each of two species in fields where those two species were detected co-occurring. The horizontal line inside the box corresponds to the median. The asterisk indicates a significant difference according to the non-parametric Kruskal-Wallis test (p < 0.05).

Composition and species diversity of whiteflies differ among Brazilian regions

The predominance of species composing the whitefly community across macroregions varied considerably. While T. acaciae predominated in the North, Southeast and Northeast, it was not detected in the Midwest (Fig. 5A). In addition, B. tuberculata was detected in all regions, and was prevalent in the South and Midwest. BtMEAM1, although not prevalent in any of the regions, was also detected in all regions. Although the number of species detected was higher in the Southeast, where six species out seven were detected, whitefly diversity was significantly higher in fields in the Northeast according to Simpson’s index of diversity (Fig. 5B), with no differences among the other four regions.

Figure 5 Composition and species diversity of whitefly populations differ among Brazilian regions.

(A) Pie charts represent the distribution of the 1,385 individuals genotyped in this study in the five geographic regions of Brazil. (B) Boxplots correspond to Simpson’s index of diversity (1-D) calculated for each geographic region. The index was first calculated for each sampled site and grouped by geographic region. Different letters indicate significant differences between groups according to the non-parametric Kruskal-Wallis test followed by post hoc multiple comparison test (p < 0.05).

No begomoviruses detected infecting cassava

To verify the presence of begomoviruses infecting cassava, we analyzed leaves sampled in some of the fields where whiteflies were collected (Table S1). Based on PCR detection using universal primers for begomoviruses, all plants were negative. Although the possibility of false negatives cannot be completely discarded, it is unlikely since the PCR assay used primers known to detect CMBs (Berrie, Rybicki & Rey, 2001; Rojas et al., 1993; Zhou et al., 1997) and none of the plants displayed symptoms (Table S1). Nevertheless, as a confirmatory step, we performed RCA followed by digestion with MspI in five samples, all with negative results (Table S1).

Discussion

Vectors play an essential role during the life cycle of plant viruses, directly affecting their ecology and evolution (Gallet, Michalakis & Blanc, 2018; Gutierrez & Zwart, 2018; Sacristan et al., 2003). Usually, a group of plant viruses establishes a very specific interaction with only one or a few related species of vectors, making virus ecology strongly dependent on that of its vector (Gallet, Michalakis & Blanc, 2018). It has been suggested that the natural host range of a virus is dependent on its vector’s host range, as most plant viruses have greater specificity for the vector than for the plant host (Dietzgen, Mann & Johnson, 2016; Elena, Fraile & Garcia-Arenal, 2014). Indeed, the existence of a competent vector for transmission and able to colonize potential reservoir and recipient new hosts is a primary ecological factor driving host range expansion of viruses. Thus, vectors play an essential role during viral disease emergence and epidemics (Elena, Fraile & Garcia-Arenal, 2014; Fereres, 2015; Gilbertson et al., 2015; Navas-Castillo, Fiallo-Olivé & Sánchez-Campos, 2011). Understanding ecological factors, such as vector species dynamics in crops, might provide important clues about historical and current events of emergence or re-emergence of viral diseases, and even anticipate the potential for new ones to occur (Legg et al., 2014).

Although it could be suggested that there are no begomoviruses capable of infecting cassava in the Americas, the high diversity of begomoviruses reported in a broad range of cultivated and non-cultivated plants in several botanical families, including the Euphorbiaceae, make this highly unlikely (Albuquerque et al., 2012a; Albuquerque et al., 2012b; Castillo-Urquiza et al., 2008; Fernandes et al., 2008; Fernandes et al., 2011; Macedo et al., 2018; Mar et al., 2017b; Paz-Carrasco et al., 2014; Rocha et al., 2013; Rodríguez-Negrete et al., 2019). Besides, CMBs could be introduced from infected rootstocks, as demonstrated by the introduction of Sri Lankan cassava mosaic virus into China (Wang et al., 2020; Wang et al., 2019). Thus, the absence of a competent vector able to colonize cassava and transfer begomoviruses from wild plants to cassava, as previously suggested (Carabali et al., 2005), seems to be a more plausible hypothesis to explain the lack of begomovirus epidemics in this crop.

Our country-wide survey of whiteflies associated with cassava in Brazil uncovered a high degree of species diversity and showed that T. acaciae and B. tuberculata are the prevalent species across the country. Non-B. tabaci species, including B. tuberculata, have been shown to be prevalent also in Colombia (Gómez-Díaz, Montoya-Lerma & Muñoz Valencia, 2019). In contrast, in Africa, endemic species of the B. tabaci complex are prevalent in cassava (Jacobson, Duffy & Sseruwagi, 2018; Legg et al., 2014; Tocko-Marabena et al., 2017). Previous studies surveying whitefly diversity in South American countries failed to detect T. acaciae and B. tuberculata in crops other than cassava, indicating a very narrow host range, which may in fact be restricted to cassava or at least to cultivated plants (Alemandri et al., 2015; Marubayashi et al., 2014; Moraes et al., 2018).

BtMEAM1 and BtNW are reported here for the first time in cassava in Brazil. BtMEAM1 was the third most prevalent species, representing 18% of the genotyped individuals, and with similar incidence to T. acaciae and B. tuberculata. The failure of previous studies to detect BtMEAM1 in cassava may have been due to the small number of samples analyzed. The wide distribution and prevalence of BtMEAM1 in the main agroecological zones in Brazil has been well established, mostly in association with annual crops such as soybean, cotton, common bean and tomato (Moraes et al., 2018). In these crops, BtMEAM1 has a great reproductive capacity, rapidly increasing its population. Interestingly, our data showed the higher prevalence of BtMEAM1 to be in the Midwest, where extensive agriculture predominates. The harvest of annual crops in the Midwest might cause the migration of the insect to semiperennial hosts such as cassava, which could explain why in some sites where BtMEAM1 predominated among adults, it was not detected as nymphs (e.g., sites MT5, MT6, PR4).

It will be important to monitor BtMEAM1 populations in cassava over the next years, to assess its possible adaptation to this host. The fact that we collected BtMEAM1 nymphs at several locations suggests that this process may already be under way. We also detected BtMED, a worrying result given the recent introduction of this species in the Brazil and its potential to displace other species, including BtMEAM1 (Liu et al., 2012; Sun et al., 2013; Watanabe et al., 2019). BtMED has disseminated quickly in the country, mainly in association with ornamental plants in greenhouses (Moraes et al., 2018). Even though we detect BtMED associated to cassava, we cannot infer its potential to effectively colonize this host since only adults were collected. The third species detected is the indigenous BtNW. Although BtMEAM1 partially displaced BtNW in Brazil, this species can still be sporadically detected, mostly in association with non-cultivated hosts (Marubayashi et al., 2014; Marubayashi et al., 2013; Moraes et al., 2018). It has been recurrently detected in Euphorbia heterophylla, suggesting a potential to colonize other species in the family Euphorbiaceae. However, the very low frequency with which it was detected and the absence of nymphs indicate that BtNW is poorly adapted to cassava.

The identification of two putative new species highlights the remarkable genetic diversity of whiteflies. Interestingly, one of the new species was collected in the state of Mato Grosso, which corresponds to the region considered to be the domestication center of cassava (Clement et al., 2016; Leotard et al., 2009; Olsen & Schaal, 1999; Watling et al., 2018). Further studies are needed to explore plant biodiversity in this region (Nassar, 2001; Olsen, 2004), which might reveal a similar diversity of whiteflies which may be specifically adapted to non-cultivated plant species due to long term co-evolution. The close phylogenetic relationship of the new species with non-B. tabaci whiteflies suggests that they are not virus vectors.

Whitefly species richness in cassava is just starting to be assessed and may be greater than reported here. Based on morphological characters, Alonso, Racca-Filho & Lima (2012) reported the presence of Aleurothrixus aepim and Trialeurodes manihoti colonizing cassava in the state of Rio de Janeiro. Although we did not analyze samples from that region, the failure to detect these species in other states suggests a restricted occurrence. Moreover, morphological characters alone are not always sufficient to classify whiteflies at the species level, and additional studies using molecular tools are needed to assess these molecularly uncharacterized whiteflies species (Dickey et al., 2015).

Host suitability has been shown to be an important factor influencing the competitive capacity among species of the B. tabaci complex (Luan et al., 2012; Sun et al., 2013; Watanabe et al., 2019). Watanabe et al. (2019) demonstrated that displacement capacity between two invasive B. tabaci species was dependent on host suitability. While BtMEAM1 displaced BtMED only on tomato, BtMED displaced BtMEAM1 on sweet pepper and common bean. Luan et al. (2012) demonstrated that even in a host plant poorly suitable for BtMEAM1, it was able to displace an indigenous species challenger. These authors demonstrated that even though host suitability may affect the speed of displacement, it may not affect the direction, as BtMEAM1 always won the challenge (Luan et al., 2012). Interestingly, two or more species occurring sympatrically were detected in 51% of the fields analyzed in our study. In sites where BtMEAM1 and B. tuberculata co-occurred, B. tuberculata predominated, suggesting a higher competitive capacity. Nonetheless, in all other combinations of co-occurring species, no differences in prevalence were observed. Thus, competitive capacity is unlikely to explain the low prevalence of BtMEAM1, or the differences observed between T. acaciae and B. tuberculata. However, we collected whiteflies in a single point in time at each location. Since competition is a dynamic process, additional surveys should be conducted in the future to further confirm these observations.

Host adaptation may be a more important component affecting the low predominance of Bt MEAM1 in cassava, as previously suggested (Carabali et al., 2005). The inability of BtMEAM1 and BtMED to colonize domesticated cassava efficiently has been demonstrated under experimental conditions (Carabali, Belloti & Montoya-Lerma, 2010; Carabali, Montoya-Lerma & Bellotti, 2008; Milenovic et al., 2019; Vyskočilová, Seal & Colvin, 2019). Carabali, Montoya-Lerma & Bellotti (2008), evaluating the colonization potential of BtMEAM1 in three commercial cassava genotypes, demonstrated that only in one of them did BtMEAM1 complete its development cycle from eggs to adult, and even then, at very low rates (0.003%). Using an electrical penetration graph assay, Milenovic et al. (2019) demonstrated the inability of BtMED to feed in cassava plants. Adults of this species spent a very short time ingesting cassava phloem sap compared to sap from a suitable host, suggesting that they would die by starvation in the field. Furthermore, the low efficiency of whiteflies of the BtMED mitochondrial subgroups Q1 and Q2 in using cassava as a host has also been demonstrated (Vyskočilová, Seal & Colvin, 2019). Oviposition and adult survival rates were very low, and development from eggs to adults was not observed. Although these studies were conducted under experimental conditions, the low predominance of BtMEAM1 and BtMED shown here and in other field surveys in Africa (Ghosh, Bouvaine & Maruthi, 2015; Tajebe et al., 2015; Tocko-Marabena et al., 2017) strongly indicates a low adaptation of these species to cassava.

Nevertheless, our results indicate an ongoing adaptation process of BtMEAM1 to cassava, with the detection of nymphs and adults in the same field. Interestingly, Carabali et al. (2005) demonstrated a gradual increase in the rate of reproduction and development of BtMEAM1 after successive passages on plants phylogenetically related to the genus Manihot (Euphorbia pulcherrima and Jatropha gossypiifolia), indicating the potential of this whitefly species to become adapted to cassava through intermediate hosts. Furthermore, successful reproduction in the wild relative M. esculenta ssp. flabellifolia indicates that this plant may constitute an intermediate host leading to adaptation (Carabali, Belloti & Montoya-Lerma, 2010). This plant has been reported to be widely spread in the Amazon basin and the Midwest region of Brazil (Olsen, 2004). Interestingly, our data showed the higher prevalence of BtMEAM1 to be in the Midwest. Although we cannot establish a cause and effect relationship, it is reasonable to speculate that M. esculenta ssp. flabellifolia could be acting as an intermediate host mediating adaptation. A survey addressing whitefly diversity in this host should be necessary to test this hypothesis.

In Brazil, cassava is predominantly grown as a subsistence crop, usually side by side with other vegetables and with a high incidence of weeds. Growing cassava in a heterogenous environment, especially in the presence of related plants, may increase the adaptation potential of BtMEAM1 and other species of the complex such as BtMED, which we also detected in the open field. A high diversity of plants in cassava fields may allow an overlapping of ecological niches for distinct whitefly species, which under enough selection pressure may gradually adapt to new hosts. The sympatric occurrence of T. acaciae, B. tuberculata and BtMEAM1, supports the role of botanical heterogeneity in shaping the composition of whitefly populations associated with cassava. A similar pattern was observed in Colombia, with 66% of the surveyed sites showing at least two species occurring sympatrically (Gómez-Díaz, Montoya-Lerma & Muñoz Valencia, 2019). Moreover, a predominance of one species in a given developmental stage and a different one in another stage (e.g., nymphs vs adults) at the same site suggests that other hosts may sustain reproduction and development, with adults migrating to cassava.

Euphorbia heterophylla (family Euphorbiaceae) is an invasive weed widely spread across Brazil and associated with several crops (Mar et al., 2017b; Wilson, 1981). The presence of E. heterophylla plants in association with cassava (Fig. 1A) and the fact that it was the most suitable host for BtMEAM1 in Brazil out seven tested (Sottoriva, Lourenção & Colombo, 2014) shows its potential to act as an intermediate host mediating BtMEAM1 adaptation. E. heterophylla has been frequently associated with the begomovirus Euphorbia yellow mosaic virus (EuYMV) (Mar et al., 2017b). Barreto et al. (2013) demonstrated that this plant is also a host of Tomato severe rugose virus (ToSRV), which even at a very low titer was transmitted to tomato plants, demonstrating the potential of E. heterophylla to act as a reservoir host. Surprisingly, considering that E. heterophylla and tomato belong to distinct botanical families, EuYMV is able to infect tomato (Barreto et al., 2013). The closer botanical relationship between E. heterophylla and cassava may indicate a higher potential of EuYMV to infect cassava. The presence of EuYMV-infected E. heterophylla in cassava fields, as observed in this study (Fig. 1A), its suitability as a host for BtMEAM1, and the high efficiency of EuYMV transmission by BtMEAM1 (Mar et al., 2017a), suggest that EuYMV may have spillover potential to cassava. Experiments are ongoing in our laboratory to assess this spillover potential.

The emergence of begomoviruses in tomato crops in Brazil followed the introduction of BtMEAM1 (Ribeiro et al., 1998; Rocha et al., 2013), demonstrating the role of vector populations in promoting viral host range expansion and consequently epidemics. Thus, the adaptation of whiteflies to cassava could facilitate the emergence of begomoviruses in this crop. The establishment of management strategies to prevent or at least delay the adaptation process is therefore necessary. Bemisia tabaci species may disperse across long distances via international trade routes (Hadjistylli, Roderick & Brown, 2016). Thus, preventing the introduction of cassava-adapted B. tabaci species from Africa should also be a priority.

Conclusions

We hypothesized that the absence of cassava-infecting begomoviruses in Brazil would be due to lack of competent B. tabaci vector species that efficiently colonize cassava. The results of our country-wide survey provide support to this hypothesis, with the most prevalent species being the non-vectors Tetraleurodes acaciae and Bemisia tuberculata. However, we did detect the presence of adult insects of Bemisia tabaci MEAM1, suggesting an ongoing adaptation process of this species to cassava which could facilitate the emergence of begomoviruses in this crop. Management strategies to prevent or at least delay the adaptation process are necessary.

Supplemental Information

Supplemental Information 1 Number and locations of samples used for begomovirus detection

Click here for additional data file.

Supplemental Information 2 Whitefly isolates obtained in this study

Click here for additional data file.

Supplemental Information 3 Pairwise identity scores between mtCOI sequences of whitefly specimens analyzed in this study

Click here for additional data file.

Supplemental Information 4 Number of nymphs and adults for the three more abundant whiteflies species

Fig. S1 . Spearman’s rank correlation coefficient analysis comparing number of nymphs and adults for the three more abundant whiteflies species. Each dot represent a sampled field. Only fields where nymphs and adults or where only one phase was observed were included in this analysis. Scatter plot showing 95% of confidence interval (light green) are shown. Total, correspond all three specie plotted together.

Click here for additional data file.

Supplemental Information 5 Sampled sites and whiteflies species detected in cassava in Brazil (raw data for Table 1)

Data set 1 (raw data). Sampled sites and whiteflies species detected in cassava in Brazil.

Click here for additional data file.

Supplemental Information 6 Restriction patters for the mtCOI gene used to genotype whiteflies specimens (raw data for Table 1 and Fig. 3)

Dataset 2 (raw data). Restriction patters for the mtCOI gene used to genotype whiteflies specimens.

Click here for additional data file.

Additional Information and Declarations

Competing Interests

Author Contributions

Data Availability

The authors declare there are no competing interests.

Cesar A.D. Xavier conceived and designed the experiments, performed the experiments, analyzed the data, prepared figures and/or tables, authored or reviewed drafts of the paper, and approved the final draft.

Angélica Maria and Vinic Nogueira conceived and designed the experiments, performed the experiments, analyzed the data, prepared figures and/or tables, and approved the final draft.

Luís Fernando Maranho Watanabe conceived and designed the experiments, performed the experiments, prepared figures and/or tables, and approved the final draft.

Tarsiane Mara Carneiro Barbosa performed the experiments, prepared figures and/or tables, authored or reviewed drafts of the paper, and approved the final draft.

Miguel Alves Júnior, Leonardo Barbosa, José E.A. Beserra-Júnior, Alessandra Boari, Renata Calegario, Eduardo Silva Gorayeb, Jaime Honorato Júnior, Gaus Silvestre de Andrade Lima, Raquel Neves de Mello, Fábio Nascimento Silva and Enilton Nascimento Santana performed the experiments, authored or reviewed drafts of the paper, and approved the final draft.

Gabriel Koch, Cristian Lopes, Késsia Pantoja, Roberto Ramos Sobrinho and José Wilson Pereira da Silva performed the experiments, prepared figures and/or tables, and approved the final draft.

Renate Krause-Sakate and Francisco M. Zerbini conceived and designed the experiments, analyzed the data, authored or reviewed drafts of the paper, and approved the final draft.

The following information was supplied regarding data availability:

The whitefly mtCOI sequences are available in GenBank: MT901081 to MT901172, MT904381, and MT904382.

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
