# Peer review of "Assessing the diversity of whiteflies infesting cassava in Brazil"

_PeerJ, doi:10.7717/peerj.11741_

## Round 0.1 · original submission · Major Revisions

Dear Dr. Xavier and colleagues:

Thanks for submitting your manuscript to PeerJ. I have now received three independent reviews of your work, and as you will see, the reviewers raised some concerns about the research (and the manuscript). Despite this, these reviewers are optimistic about your work and the potential impact it will have on research studying the whitefly-begomovirus-cassava system. Thus, I encourage you to revise your manuscript, accordingly, taking into account all of the concerns raised by both reviewers.

While the concerns of the reviewers are relatively minor, this is a major revision to ensure that the original reviewers have a chance to evaluate your responses to their concerns. There are many suggestions, which I am sure will greatly improve your manuscript once addressed.

Please ensure the relevant literature is cited, and that alternative vectors and systems are mentioned. The suggestions on methodological improvements by the reviewers should be addressed.

Please note that reviewer 3 has included a marked-up version of your manuscript.

Therefore, I am recommending that you revise your manuscript, accordingly, taking into account all of the issues raised by the reviewers.

I look forward to seeing your revision, and thanks again for submitting your work to PeerJ.

Good luck with your revision,

-joe

Reviewer 1 ·

Basic reporting

Cassava (Manihot esculenta Crantz, Family Euphorbiaceae) is a tropical food crop for about a fifth of the world’s population, and one of the most efficient crops for converting solar energy to produce carbohydrates. Cassava mosaic geminiviruses (CMVs) cause cassava mosaic disease (CMD). Beside of where they origninated from African, in the last few years, there are a few reports about CMV expanding invasive infection in new counties of SouthEast Asia and East Asia, including Cambodia,Vietnam, China and so on ( Wang HL et al. Plant Disease 2016, Wang D et al. Plant Disease 2018,Minato et al. PLoS One 2019). CMV not also could infect cassava, but also jump into new host such as Jatropha curcas, an euphorbiaceae biofuel crop (Gao et al. Arch Virology 2010, Wang et al. Virus genes 2014). Therefore it is important to surveillance for invasive CMV in new world such as Southern Amercian, eg. Brazil. This work has its importance to be published in PeerJ. I suggest they include recent references listed above. Esp. the fact that there are at least two CMVs detected in China (Wang D et al. Plant Disease 2018, Wang D et al. Phytopathology Research 2020) are thought to be root-stock originated from Cambodia. The authors may discuss alternative transmission way instead of whitefly vector, but rootstock and international importing. They further confirmed that scucess virus transmission from the symptomatic cassava by viruliferous whiteflies Bemisia tabaci MEAM1 (Middle East-inor 1, formerly biotype B) into tomato (Solanum lycopersicum, Moneymaker) or Arabidopsis thaliana (Col-0). Importantly, both of two groups failed to detect any geminivirus with the universal primers PA/PB (PA,5’-TAATATTACCKGWKGVCCSC-3’; PB, 5’-TGGACYTTRCAWGGBCCTTCACA-3’) (Deng et al. 1994) on symptomatic cassava, suggesting the tranditional primers and method may not be suitable for detect all geminiviruses. Wang D et al. Phytopathology Research 2020, provided an improved primers for rapid detection method.

Experimental design

The authors should tune down their claims and also better to discuss alternative explainations on their results based on several published researches.

Validity of the findings

They'd better to use optimized primers developed in Wang D et al. Phytopathology Research 2020, for rapid detection method.

Additional comments

It is a very important survey experiment. The authors should be noticed with published new data about recent CMVs invasion events and be very careful with their data interpretation about the conclusion of no presence of any geminivirus in Brazil.

·

Basic reporting

The authros reported a country-wide whitefly diversity study in cassava in Brazil. A large number of whitefly specimens were collected from diversed areas and characterised by mitochondrial cytochrome oxidase subunit I sequening. The study updates the status of whitefly diversity in Brazil and hence it is important. The manuscript is well written, methodology is adequate, results are well presented. I suggest acceptance of the manuscript with minor revision.

Experimental design

From the methodology section, it seems that the PCR from individual whitefly DNA was carried out and sequenced. The utility of fRFLP analsysis of the amplified PCR products is not clear as the current method of whitefly genotyping is sequence-specific. Please clarify this section.
Lines 206-209- difficult to follow. Were the additional individuals for each stage analyzed in RFLP or sequenced?
The authors have not detected the presence of begomovirus in the cassava plants. If there are some other begomovirus infected plants nearby, have the author checked the presence of begomoviruses in the vector? I believe by doing so, the authors can compare if the vectors are incapable of transmitting the virus or the plants are resistant to the nearby begomoviruses.
The authors used universal DNA primers for begomovirus detection and no begomovirus was detected. I suggest to undertake a rolling cycle amplification and digestion as a confirmatory step.

Validity of the findings

I suggest the following revisions:
As there is no detetction of begomoviruses and I did not find any evidence in the current study that shows the begomovirus dynamics, the title may be revised and restricted to the whitefly diversity in cassava.
The interpretation of RFLP data is not adequate.
The authors stated the specimens were collected from 66 spots of different agroecological zones of Brazil. I suggest to provide a figure that shows the locations on the map.
Diversity of the whitefly in each spots will also provide important data on their distribution.
The altitude, temperature and time of collection would provide important information on their ecological and host adaptation. This should be included in the revised version.

Additional comments

B. tabaci MEAM1 has been recorded by the authors. Whether it can transmit the begomoviruses infecting other crops at the same ecological niches? or the cassava cultivars are resistant to begomoviruses: will be an interesting study. B. tabaci MEAM1 is a superior vector of many begomovirus species. But this is not universal. The efficiency of other predominat B tabaci species can also be checked for a comprehensive understanding.

Reviewer 3 ·

Basic reporting

The authors mentioned cassava virus disease such as CMD was severe in Africa. However, they did not listed the key competent vector, i.e., the cryptic species of Bemisia tabaci complex that transmit the begomoviruses in Africa. Is the BtMEAM1 the competent vector for CMD or other indigenous whitefly the competent vector in Africa? This information should be provided for readers as it is helpful to infer the potential begomovirus occurrence in South America.

Experimental design

no comment

Validity of the findings

no comment

Additional comments

Based on the phenomenon that begomoviruses were not occurred in cassava in South America, this manuscript investigates whitefly diversity, composition and prevalence in Brazil. The hypothesis proposed by authors is that begomoviruses epidemic have not occurred in cassava due to lack of competent vectors. This is an interesting topic with valuable information for vector and disease interaction and cassava production.

Annotated reviews are not available for download in order to protect the identity of reviewers who chose to remain anonymous.

---

## Round 0.2 · Minor Revisions

Dear Dr. Xavier and colleagues:

Thanks for revising your manuscript. The reviewer is generally satisfied with your revision (as am I). Great! However, there is a request to alter your title. Please attend to this issues ASAP so we may move towards acceptance of your work.

Best,

-joe

·

Basic reporting

Adequate

Experimental design

Adequate

Validity of the findings

Adequate

Additional comments

The authors have not detected any begomovirus during the study. The occurrence or dynamics of begomoviruses have not been reported here. I suggest revising the title to "Assessing diversity of whitefly infesting cassava in Brazil"

---

## Round 0.3 · accepted · Accept

Dear Dr. Xavier and colleagues:

Thanks for revising your manuscript. I now believe that your manuscript is suitable for publication. Congratulations! I look forward to seeing this work in print, and I anticipate it being an important resource for groups studying the whitefly-begomovirus-cassava system. Thanks again for choosing PeerJ to publish such important work.

Best,

-joe